# Common adolescent mental health disorders seen in Family Medicine Clinics in Ghana and Nigeria

Sonny John Kumbet[1], Tijani Idris Ahmad Oseni [2,3]*, Magdalene Mensah-Bonsu[4], Fatima Mohammed Damagum[5], Edwina Beryl Addo Opare-Lokko[6,7], Eve Namisango[8,9‡], AbdulGafar Lekan Olawumi[5‡], Onyenwe Chibuike Ephraim[10], Benjamin Aweh[11]

1 Department of Family Medicine, Geriatric Medicine Sub Division, Federal Medical Centre, Keffi, Nigeria, 2 Department of Family Medicine, Ambrose Alli University, Ekpoma, Nigeria, 3 Lifestyle and Behavioural Medicine Unit, Department of Family Medicine, Irrua Specialist Teaching Hospital, Irrua, Nigeria, 4 Department of Family Medicine, Komfo Anokye Teaching Hospital, Kumasi, Ghana, 5 Department of Family Medicine, Aminu Kano Teaching Hospital, Kano, Nigeria, 6 Department of Family Medicine, Greater Accra Regional Hospital, Accra, Ghana, 7 Faculty of Family Medicine, Ghana College of Physicians and Surgeons, Accra, Ghana, 8 African Palliative Care Association, Kampala, Uganda, 9 College of Health Sciences, Makerere University, Kampala, Uganda, 10 Department of Family Medicine, Alex Ekwueme Federal University Teaching Hospital, Abakaliki, Nigeria, 11 Department of Mental Health and Behavioural Medicine, Irrua Specialist Teaching Hospital, Irrua, Nigeria

☯ These authors contributed equally to this work.
‡ EN and ALO also contributed equally to this work.
* tijanioseni@aauekpoma.edu.ng

**Data Availability Statement:** All relevant data are within the manuscript and its Supporting Information files.

## Abstract

### Background

Mental health disorders among adolescents is on the rise globally. Patients seldom present to mental health physicians, for fear of stigmatization, and due to the dearth of mental health physicians. They are mostly picked during consultations with Family Physicians. This study seeks to identify the common mental health disorders seen by family Physicians in Family Medicine Clinics in Nigeria and Ghana.

### Methods

A descriptive cross-sectional study involving 302 Physicians practicing in Family Medicine Clinics in Nigeria and Ghana, who were randomly selected for the study. Data were collected using self-administered semi-structured questionnaire, and were entered into excel spreadsheet before analysing with IBM-SPSS version 22. Descriptive statistics using frequencies and percentages was used to describe variables.

### Results

Of the 302 Physicians recruited for the study, only 233 completed the study, in which 168 (72.1%) practiced in Nigeria and 65 (27.9%) in Ghana. They were mostly in urban communities (77.3%) and tertiary health facilities (65.2%). Over 90% of Family Medicine practitioners attended to adolescents with mental health issues with over 70% of them seeing at least 2

**Funding:** The author(s) received no specific funding for this work.

**Competing interests:** The authors have declared that no competing interests exist.

adolescents with mental health issues every year. The burden of mental health disorder was 16% and the common mental health disorders seen were depression (59.2%), Bipolar Affective Disorder (55.8%), Epilepsy (51.9%) and Substance Abuse Disorder (44.2%).

## Conclusion

Family Physicians in Nigeria and Ghana attend to a good number of adolescents with mental health disorders in their clinics. There is the need for Family Physicians to have specialized training and retraining to be able to recognize and treat adolescent mental health disorders. This will help to reduce stigmatization and improve the management of the disease thus, reducing the burden.

## Introduction

The burden of mental health disorders is on the rise globally. In 2019, one in every eight people, or 970 million people around the world were living with a mental disorder, in which anxiety and depressive disorders were the most common [1]. In 2020, the amount of people living with anxiety and depressive disorders rose considerably because of the COVID-19 pandemic [1, 2].

Adolescents are group of persons with chronological age 10–19 years of age. Adolescence is a critically important stage of life for mental health and well-being of individuals, not only for the reason that this is when young people acquire autonomy, social interaction, self-control, and rapid learning, but also because the abilities and potentials formed in this period have a direct bearing on their mental health for the rest of their lives [3]. Although adolescents generally are highly susceptible to mental health challenges, they receive very little attention, especially in developing countries [4]. Globally, one in seven adolescents experience a mental disorder, accounting for 13% of the global burden of disease in this age group. Depression, anxiety and behavioural disorders are among the leading causes of impairment and disability among them [5].

According to the World Health Organization, common mental disorders such as depression and anxiety account for the largest proportion of mental, developmental and substance use disorders.

Behavioural disorders including attention-deficit and hyperactivity disorder plus conduct disorder are more prevalent among 10-14-year olds, while alcohol and drug use disorders more common in older adolescence (15-19-year olds) [3].

One out of every six young Nigerian aged 15–24 is suffering from poor mental health, according to a report released by the United Nations Children Fund (UNICEF) [6]. In Ghana, WHO estimated that 650,000 are suffering from severe mental disorder and a further 2,166,000 are suffering from moderate to mild mental disorder with a treatment gap of 98% from the total population [7]. However, the prevalence of mental illness and its burden among adolescents is not known at the national level [8].

There is a dearth of mental health experts in West Africa. This is worsened by the stigma associated with mental health disorders in the region. The World Health Organization (WHO) and World Organization of Family Doctors (WONCA) advocates for the integration of mental health services into primary care as the most viable way of closing the treatment gap and ensuring people get the mental health care they need [9]. There is the need to ascertain the degree of integration of mental health into primary care in Nigeria and Ghana. This can be

accessed by evaluating how commonly primary care physicians in these two countries see adolescents with mental health disorders. This will help policy makers identify the role of Family Physicians in the management of adolescent mental health disorders. It will also help identify the common mental health disorders presenting to the family medicine clinics. This could be used to increase capacity and training of family physicians in the management of adolescent mental health disorders and will on the long run reduce the burden of the disease which is currently high. The aim of this study was to identify the common mental health disorders among adolescents seen by family Physicians in Family Medicine Clinics in Nigeria and Ghana.

## Materials and methods

### Study design and setting

The study was a descriptive cross-sectional study conducted among 302 Family Physicians practising in Nigeria and Ghana. It is part of a larger study with some of the findings already published [10]. The sample size was calculated to be 302 using fisher's formula and finite correction was made based on the total population of Family Physicians in each country: 1200 and 125 for Nigeria and Ghana respectively. The sample size (302) was proportionately distributed to each country based on their population size at the time of data collection: Nigeria had 254 while Ghana had 48 Family Physicians. The study sites included General Outpatient Clinics of Teaching Hospitals, Specialist, General and District Hospitals and other Primary Healthcare Centers, where Family Physicians practice in both countries. Physicians were recruited into the study using multi-stage sampling methods. Purpose sampling was used to select two countries, Nigeria and Ghana for the study. Family Medicine clinics in both countries were identified through the Family Medicine accreditation lists of the West African College of Physicians, the postgraduate medical colleges of Nigeria and Ghana, and the Society of family Physicians of Nigeria and Ghana's database. Simple random sampling was then used to select Family Medicine clinics across both countries from this database and all the Family Physicians in the selected clinics that met the selection criteria and gave written informed consent were recruited for the study. The questionnaire was in two parts: 1) Sociodemographic variables of the physicians themselves and 2) Data obtained from the family physicians' clinic over six weeks. The physicians reported the number of mental health conditions seen in adolescents over the past six weeks from their clinical records. The medical records of the family medicine clinics of patients seen were reviewed by the physicians to confirm the number and distribution of adolescent mental health patients they have attended to and reported accordingly. Other details of the study design and methodology are contained in the published article [10].

### Statistical analysis

Data were analysed using the Statistical Package for Social Sciences™ (IBM Corp, Armonk, NY, USA) version 22.0. They were presented in tables and were described using frequencies and percentages.

## Results

A total of 233 Family Physicians completed the study (77.2% response rate with a country response rate for Nigeria and Ghana of 66.1% and 135% respectively), in which 65 (27.9%) in Ghana and 168 (72.10%) in Nigeria participated in the study. They worked in facilities that were mainly in the urban setting 180 (77.25%). Majority of the facilities were Tertiary institutions 152 (65.24%), which was either Teaching Hospitals or Federal Medical Centres.

The socio-demographic characteristics are shown in Table 1.

**Table 1. Socio-demographic characteristics of respondents (N = 233).**

| VARIABLE | FREQUENCY | PERCENTAGE |
|---|---|---|
| **Country** | | |
| Ghana | 65 | 27.90 |
| Nigeria | 168 | 72.10 |
| **Location of Health Facility** | | |
| Rural | 53 | 22.75 |
| Urban | 180 | 77.25 |
| **Type of Facility** | | |
| General/District Hospital | 55 | 23.61 |
| Teaching Hospital/Federal Medical Centre | 152 | 65.24 |
| Private/Mission Hospital | 26 | 11.15 |

Table 2 shows the adolescent mental health disorders seen in Family Medicine Clinics in Ghana and Nigeria. Over 90% of Family Medicine practitioners attend to adolescents with mental health issues with over 70% of them seeing at least 2 adolescents with mental health issues every year. The burden of adolescent mental health disorders seen by Family Physicians (> 3 patients a year) in this study was 16%.

The distribution of the common adolescent mental health disorders seen in Family Medicine Clinics in Ghana and Nigeria are as shown in Table 3. Depression 138 (59.23%) was the most commonly seen disorder followed by Bipolar Disorders 130 (55.79%), and Substance Use Disorders 103 (44.21%) in that order.

## Discussion

The study participants and their distribution have already been described in another publication from the study [10]. Our study showed that 91% of respondents attend to adolescents with mental health issues with over half of them attending to about two to three adolescents with mental health disorders yearly. This is worrisome, it shows that the burden of adolescent mental health in primary care is enormous. If not attended to, it usually leads to major mental problems including suicide and self-harm which have been on the rise among adolescents [11, 12]. It calls for improved capacity in the diagnosis and management of mental health disorders among Family Physicians. This is particularly so as patients hardly present for care as they are generally stigmatized, shunned and denied access to care by their families, caregivers and the society [13]. The burden of mental health disorders among adolescents in primary care in west Africa in this study (16%) is similar to the Global findings of 14% burden of mental health disorders among adolescents [5] and 16% burden found by Robert et al in England [14]. The high burden found in this study can be attributed to the large number of adolescents presenting to

**Table 2. Adolescent mental health disorders seen in primary care clinics (N = 233).**

| VARIABLE | FREQUENCY | PERCENTAGE |
|---|---|---|
| **Do you attend to adolescents with mental health issues? (n = 233)** | | |
| Yes | 212 | 90.99 |
| No | 21 | 9.01 |
| **How often do you attend to adolescents with mental health issues? (n = 212)** | | |
| Often (> 3 patients a year) | 33 | 15.57 |
| Sometimes (2–3 patients a year) | 117 | 55.19 |
| Rarely (≤ 1 patient a year) | 62 | 29.24 |

Table 3. Common adolescent mental disorders seen in primary care clinics (N = 233).

| DISORDER | FREQUENCY | PERCENTAGE |
| --- | --- | --- |
| Depression | 138 | 59.23 |
| Substance Use Disorders | 103 | 44.21 |
| Bipolar Disorders | 130 | 55.79 |
| Psychosis | 52 | 22.32 |
| Suicide or Self Harm | 71 | 30.47 |
| Schizoaffective Disorder | 58 | 24.89 |
| ADHD | 39 | 16.74 |

primary care centres as compared to specialist clinics, the fear of stigmatization of mental health disorders and the recent development of subspecialty clinics such as adolescent clinics, and geriatric clinics in General outpatient clinics [3, 8].

Respondents reported a variety of mental health disorders seen. The most common disorder was depression. This was followed by bipolar disorder, epilepsy and substance use disorders with the least common disorders as enuresis, Attention Deficit Hyperactivity Disorder (ADHD), Psychosis and Schizoaffective Disorders. World Health Organization (WHO) report that 12 billion work hours and 1 trillion US dollars are lost annually to depression and anxiety alone [15]. On a global scale WHO stated in 2021 that "Depression, anxiety and behavioural disorders are among the leading causes of illness and disability among adolescents" [16], which is in tandem with this study having 59.23% of the respondents treating depression. However, a study done in Enugu, Nigeria [17] had Schizophrenia as the commonest mental health disorder in Nigeria. The above study was not done in primary care setting and Providers and stakeholders had limited or no training in adolescent mental health and that could be the reason for the slight difference. The high number of depressions among adolescent can be explained by the high level of poverty and few numbers of specialist to attend to the huge burden of mental health disorder among adolescents [3].

Bipolar disorder was the second leading mental health disorder identified in this study. This could be due to the fact that the most frequent range of onset of bipolar disorder is between the ages of 14–21 years; which falls with the adolescent and early adult age group [18].

Substance use disorder and suicide or self-harm were also prevalent among adolescent in this study. This is similar to the findings of Birhanu *et al* in Ethiopia [19], Mavura *et al* in northern Tanzania [20], and Volkow *et al* in the US [21]. The reasons may not be unconnected to the high level of peer influence, risk taking behaviour and experimentation with substances due to developmental changes and challenges in adolescence [19, 20]. The high burden of self-harm or suicide in this study could be due to the strong relationship between substance abuse and suicide or self-harm especially, among adolescents and young adults [22, 23].

There is an urgent need for Family Physicians to look out for adolescent mental health issues and address them at the early stage before they progress to more complicated forms. There is also the need for policy makers to increase awareness on the burden of mental health disorders among adolescents and put measures in place to mitigate them.

## Conclusion

The prevalence of mental disorders among adolescents seen by Family Physicians in Family Medicine clinics are high in Nigeria and Ghana. There is the need for Family Physicians to have specialized training and retraining on mental health issues concerning adolescents. There is also a need to have more subspecialty adolescent clinics in Family Medicine Clinics to be

able to handle adolescents' challenges including recognising and treating adolescent mental health disorders easily. There is also the need for high index of suspicion for these disorders in adolescents when they present. Future studies should seek to establish the relationship between specific variables and types of mental health conditions as well as to establish the background knowledge of health workers in recognition of these conditions. Policy makers should also put measures in place to improve awareness and care for patients.

## Limitations

The study was conducted in two countries in West Africa. Though most Family Medicine Clinics in the region are in these two countries, the results still may not be a true representation of the entire region. Also, the study was conducted among doctors. However, relatively most primary health care centres in the region are run by primary care nurses, community health officers and community health extension workers. These categories of primary care providers were not included in the study even though they attend to most of the patients presenting to primary care facilities in these regions. There was an overall response rate of 77.2% while the country response rate for Nigeria and Ghana was 66.1% and 135% respectively. Being an online survey could explain the high non-response rate of 22.8%. A greater response was obtained from Ghanaian physicians. We thus recruited more physicians from Ghana to increase the power and compensate to some extent for the poor response from Nigeria.

## Supporting information

**S1 Checklist. STROBE statement—checklist of items that should be included in reports of observational studies.**
(DOCX)

**S1 Data.**
(XLSX)

## Acknowledgments

We thank Afriwon Research Group ARG of Afriwon Renaissance for bringing young researchers in Africa together and providing the platform from which the research was conducted.

## Author Contributions

**Conceptualization:** Sonny John Kumbet, Tijani Idris Ahmad Oseni, Magdalene Mensah-Bonsu, Fatima Mohammed Damagum, Edwina Beryl Addo Opare-Lokko, Onyenwe Chibuike Ephraim.

**Formal analysis:** Eve Namisango.

**Investigation:** Sonny John Kumbet, Tijani Idris Ahmad Oseni, Magdalene Mensah-Bonsu, Fatima Mohammed Damagum.

**Methodology:** Sonny John Kumbet, Tijani Idris Ahmad Oseni, Magdalene Mensah-Bonsu, Fatima Mohammed Damagum, Edwina Beryl Addo Opare-Lokko, Onyenwe Chibuike Ephraim.

**Supervision:** Tijani Idris Ahmad Oseni, Edwina Beryl Addo Opare-Lokko.

**Validation:** Sonny John Kumbet, Tijani Idris Ahmad Oseni, Magdalene Mensah-Bonsu, Fatima Mohammed Damagum, Edwina Beryl Addo Opare-Lokko, Eve Namisango, AbdulGafar Lekan Olawumi, Onyenwe Chibuike Ephraim, Benjamin Aweh.

**Writing – original draft:** Sonny John Kumbet, Tijani Idris Ahmad Oseni, Magdalene Mensah-Bonsu, Fatima Mohammed Damagum, Edwina Beryl Addo Opare-Lokko, Eve Namisango, AbdulGafar Lekan Olawumi, Onyenwe Chibuike Ephraim, Benjamin Aweh.

**Writing – review & editing:** Sonny John Kumbet, Tijani Idris Ahmad Oseni, Magdalene Mensah-Bonsu, Fatima Mohammed Damagum, Edwina Beryl Addo Opare-Lokko, Eve Namisango, AbdulGafar Lekan Olawumi, Onyenwe Chibuike Ephraim, Benjamin Aweh.

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
