## [Decision Letter · Decision Letter 0]

5 Jul 2023

PONE-D-23-13066COMMON ADOLESCENT MENTAL HEALTH DISORDERS SEEN IN FAMILY MEDICINE CLINICS IN GHANA AND NIGERIAPLOS ONE

Dear Dr. Oseni,

Thank you for submitting your manuscript to PLOS ONE. After careful consideration, we feel that it has merit but does not fully meet PLOS ONE’s publication criteria as it currently stands. Therefore, we invite you to submit a revised version of the manuscript that addresses the points raised during the review process.

We look forward to receiving your revised manuscript.

Kind regards,

Nicholas Aderinto Oluwaseyi

Academic Editor

PLOS ONE

Journal Requirements:

- https://doi.org/10.1177/11786329231166366

- https://www.who.int/europe/home?v=welcome

- https://www.who.int/news-room/fact-sheets/detail/mental-disorders

- https://search.yahoo.com/reviews?age=1m&ei=UTF-8&fr2=&p=What+is+the+average+age+for+mental+illness+to+start%3F&q=Emomali+Rahmon&v_t=rs-bot

In your revision ensure you cite all your sources (including your own works), and quote or rephrase any duplicated text outside the methods section. Further consideration is dependent on these concerns being addressed.

Reviewers' comments:

Reviewer's Responses to Questions

**Comments to the Author**

1. Is the manuscript technically sound, and do the data support the conclusions?

Reviewer #1: No

Reviewer #2: No

2. Has the statistical analysis been performed appropriately and rigorously? 

Reviewer #1: No

Reviewer #2: Yes

3. Have the authors made all data underlying the findings in their manuscript fully available?

Reviewer #1: No

Reviewer #2: No

4. Is the manuscript presented in an intelligible fashion and written in standard English?

Reviewer #1: No

Reviewer #2: Yes

5. Review Comments to the Author

Reviewer #1: COMMON ADOLESCENT MENTAL HEALTH DISORDERS SEEN IN FAMILY MEDICINE CLINICS IN GHANA AND NIGERIA

General comments: Overall I commend the authors for trying to explore the mental health disorders among adolescents- a vulnerable group of people.

Introduction: Written, however the aims of this study is at variance what was done

Methods: This method can not answer the aims of this study. The study intends to look at mental health disorders among adolescents, yet no adolescent was samples.

Estimated minimum sample size was 302, yet 233 was recruited. Thus, no conclusion can be derived since the analysed sample was not up to the minimum was set that can allowed for conclusion? By the way, what was the power and precision of the study.

A validated questionnaire? No link to this including the previous published work for verification?

“The burden of mental health disorder in this study was 16% and the distribution of the common

adolescent mental health disorders seen in Family Medicine Clinics in Ghana and Nigeria are as

shown in Table 3”- how was this burden determined, among whom, a snap shot of family physician based on recall? To diagnosis mental disorder of adolescents? Without sampling the adolescents using a standard instrument?

“Depression 138 (59.23%) was the most commonly seen disorder followed by

Bipolar Disorders 130 (55.79%), Epilepsy 121 (51.93%), and substance use disorders 103

(44.21%) in that order” -again these are categories of mental disorder with clear guideline for diagnosis/criteria to be fulfilled- check DSM V. Again, this was arrived at based on snap shot respond of family physicians on behalf of adolescents??? This study cannot answer questions on details categories of mental disorders among adolescents, when they themselves were excluded.

The authors urge to have administered standard and validated tools on the adolescents to answers their research questions.

Table 3-how was the diagnosis arrived-based on report of family physicians/hospital records-what of co-existence of more than one conditions.

Conclusion

This research method cannot support the conclusion

Recommendations: Rejected.

Reviewer #2: The manuscript is well written with a clearly stated objective which was significantly addressed. Following the review, it is believed that the manuscript will benefit greatly from a thorough language proofreading. There is a need to be concise with sentence construct throughout the article.

The objective stated that the study seeks to "evaluate" the common mental health disorders seen...This used of the quoted word should be revised as the mental health disorders were mainly identified.

Some abbreviations were used in the manuscript which were not specifically defined e.g WHO, PTSD.

There is a need to ensure that adequate information is provided in "Materials and Methods" section e.g information about the contents (sections) in the data collection tool, how the tools was deployed, validated and the parameters used for the sample size calculation. The multi-stage sampling method should be clearly explained.

Was ethical approval obtained in Ghana? If no, why?

A non-response rate over 20% was recorded. What was the reason for this? Can it be included in the limitation.

In the results section, a chart showing the common adolescent mental health disorder (by country or location of health facility) should be considered (e.g a clustered column bar chart).

What further research suggestion on this subject do the authors think should be prioritized? This could be included in the conclusion section.

6. PLOS authors have the option to publish the peer review history of their article (what does this mean?). If published, this will include your full peer review and any attached files.

Reviewer #1: No

Reviewer #2: No

---

## [Author Response · Author response to Decision Letter 0]

24 Jul 2023

The manuscript has been revised and a point by point revision attached. Thank you for your comments and hoping for a favourable review of the revised manuscript.

---

## [Decision Letter · Decision Letter 1]

2 Oct 2023

PONE-D-23-13066R1COMMON ADOLESCENT MENTAL HEALTH DISORDERS SEEN IN FAMILY MEDICINE CLINICS IN GHANA AND NIGERIAPLOS ONE

Dear Dr. Oseni,

Thank you for submitting your manuscript to PLOS ONE. After careful consideration, we feel that it has merit but does not fully meet PLOS ONE’s publication criteria as it currently stands. Therefore, we invite you to submit a revised version of the manuscript that addresses the points raised during the review process.

We look forward to receiving your revised manuscript.

Kind regards,

Nicholas Aderinto Oluwaseyi

Academic Editor

PLOS ONE

Reviewers' comments:

Reviewer's Responses to Questions

**Comments to the Author**

1. If the authors have adequately addressed your comments raised in a previous round of review and you feel that this manuscript is now acceptable for publication, you may indicate that here to bypass the “Comments to the Author” section, enter your conflict of interest statement in the “Confidential to Editor” section, and submit your "Accept" recommendation.

Reviewer #3: (No Response)

Reviewer #4: (No Response)

2. Is the manuscript technically sound, and do the data support the conclusions?

Reviewer #3: Partly

Reviewer #4: Partly

3. Has the statistical analysis been performed appropriately and rigorously? 

Reviewer #3: Yes

Reviewer #4: Yes

4. Have the authors made all data underlying the findings in their manuscript fully available?

Reviewer #3: Yes

Reviewer #4: Yes

5. Is the manuscript presented in an intelligible fashion and written in standard English?

Reviewer #3: Yes

Reviewer #4: Yes

6. Review Comments to the Author

Reviewer #3: 1. The method used to determine the 16% burden in this study needs further clarification. It would be beneficial if you could provide a more detailed explanation of how this percentage was calculated.

2. A significant disparity exists between the calculated sample size requirement and the actual distribution of participants between Ghana and Nigeria. Ghana appears to be overrepresented, while Nigeria is notably underrepresented. Moreover, the study's limitation, which attributes this deviation to being an online survey, is a critical concern that could potentially compromise the integrity of the study's conclusions. I strongly suggest further elaboration on this limitation to offer a more comprehensive assessment of its impact on the drawn conclusions.

3. Epilepsy, also known as a seizure disorder, is classified as a neurological disease rather than a mental health illness. While it's acknowledged that epilepsy can predispose individuals to mental health issues, it remains essential to adhere to accurate classification. This raises questions about the criteria used by physicians to diagnose mental health illnesses within the dataset, particularly since it's listed as the third condition with a frequency of 121 (51.93%). Clarification of the diagnostic criteria for mental health illnesses in this context would be valuable.

Reviewer #4: The statement of the problem needs to be included to the background of the abstract. For instance, what problems could be associated with patients not presenting themselves to mental health physicians? The justification for the study should also be highlighted.

Greater clarity is required in the methodology section:

- More detail on how the multistage sampling was conducted should be provided

-How were the clinic records reviewed to give information about the mental health conditions? This information must be provided?

-Even though the authors have indicated that this is part of a larger study, there is still a need for this paper to be able to stand alone and as such there must be at least some information about how the mental disorders examined in this study were assessed e.g. the instruments used to assess each of them and cutoff values. Are these instruments that can be used by any physician or is specialist training required?

Results

-Provide response rate for each country, not only the overall response rate. This would be derived from the proportionate allocation to Ghana and Nigeria respectively.

7. PLOS authors have the option to publish the peer review history of their article (what does this mean?). If published, this will include your full peer review and any attached files.

Reviewer #3: No

Reviewer #4: No

---

## [Author Response · Author response to Decision Letter 1]

5 Oct 2023

Response to reviewers

Reviewer Number Original comments of the reviewer Reply by the author(s) Changes done on page number and line number

Reviewer 3 1. The method used to determine the 16% burden in this study needs further clarification. It would be beneficial if you could provide a more detailed explanation of how this percentage was calculated. Clarification provided Page 7,

Line 138

 2. A significant disparity exists between the calculated sample size requirement and the actual distribution of participants between Ghana and Nigeria. Ghana appears to be overrepresented, while Nigeria is notably underrepresented. Moreover, the study's limitation, which attributes this deviation to being an online survey, is a critical concern that could potentially compromise the integrity of the study's conclusions. I strongly suggest further elaboration on this limitation to offer a more comprehensive assessment of its impact on the drawn conclusions. Reason elaborated more in the limitation Page 11,

Line 221;

Page 12,

Line 223

 3. Epilepsy, also known as a seizure disorder, is classified as a neurological disease rather than a mental health illness. While it's acknowledged that epilepsy can predispose individuals to mental health issues, it remains essential to adhere to accurate classification. This raises questions about the criteria used by physicians to diagnose mental health illnesses within the dataset, particularly since it's listed as the third condition with a frequency of 121 (51.93%). Clarification of the diagnostic criteria for mental health illnesses in this context would be valuable. Thank you. Epilepsy removed as it is a neurological disorder as highlighted by the reviewer. 

It was initially added based on findings of a study on mental and neurological disorders in Ghana. Page 8,

Line 145, 147 

(Table 3);

Page 9,

Line 174;

Page 10,

Line 183

Reviewer 4 - The statement of the problem needs to be included to the background of the abstract. For instance, what problems could be associated with patients not presenting themselves to mental health physicians? Included in the abstract Page 2,

Line 29

 The justification for the study should also be highlighted. Justification highlighted. Page 4,

Line 83

 Greater clarity is required in the methodology section:

- More detail on how the multistage sampling was conducted should be provided Details of the multistage sampling provided Page 5,

Line 104

 -How were the clinic records reviewed to give information about the mental health conditions? This information must be provided? The records were reviewed by the physicians to confirm the no of mental health patients they have attended to. Page 6,

Line 113

 -Even though the authors have indicated that this is part of a larger study, there is still a need for this paper to be able to stand alone and as such there must be at least some information about how the mental disorders examined in this study were assessed e.g. the instruments used to assess each of them and cutoff values. Are these instruments that can be used by any physician or is specialist training required? Done Page 5,

Line 104;

Page 6,

Line 113

 Results

-Provide response rate for each country, not only the overall response rate. This would be derived from the proportionate allocation to Ghana and Nigeria respectively. Country response rate included Page 6,

Line 126

---

## [Decision Letter · Decision Letter 2]

31 Oct 2023

COMMON ADOLESCENT MENTAL HEALTH DISORDERS SEEN IN FAMILY MEDICINE CLINICS IN GHANA AND NIGERIA

PONE-D-23-13066R2

Dear Dr. Oseni,

We’re pleased to inform you that your manuscript has been judged scientifically suitable for publication and will be formally accepted for publication once it meets all outstanding technical requirements.

Kind regards,

Nicholas Aderinto Oluwaseyi

Academic Editor

PLOS ONE

Additional Editor Comments (optional):

Reviewers' comments:

Reviewer's Responses to Questions

**Comments to the Author**

1. If the authors have adequately addressed your comments raised in a previous round of review and you feel that this manuscript is now acceptable for publication, you may indicate that here to bypass the “Comments to the Author” section, enter your conflict of interest statement in the “Confidential to Editor” section, and submit your "Accept" recommendation.

Reviewer #3: All comments have been addressed

2. Is the manuscript technically sound, and do the data support the conclusions?

Reviewer #3: Yes

3. Has the statistical analysis been performed appropriately and rigorously? 

Reviewer #3: Yes

4. Have the authors made all data underlying the findings in their manuscript fully available?

Reviewer #3: Yes

5. Is the manuscript presented in an intelligible fashion and written in standard English?

Reviewer #3: Yes

6. Review Comments to the Author

Reviewer #3: (No Response)

7. PLOS authors have the option to publish the peer review history of their article (what does this mean?). If published, this will include your full peer review and any attached files.

Reviewer #3: No

---

## [Editor Report · Acceptance letter]

7 Nov 2023

PONE-D-23-13066R2 

Common Adolescent Mental Health Disorders Seen in Family Medicine Clinics in Ghana and Nigeria 

Dear Dr. Oseni:

I'm pleased to inform you that your manuscript has been deemed suitable for publication in PLOS ONE. Congratulations! Your manuscript is now with our production department. 

Kind regards, 

on behalf of

Dr. Nicholas Aderinto Oluwaseyi 

Academic Editor

PLOS ONE